# From Change to Transformation: Living Synodality in Ministry with Young Adults

Tracey Lamont 

Loyola Institute for Ministry, Loyola University New Orleans, New Orleans, LA 70118, USA; tlamont@loyno.edu

**Abstract:** Pope Francis is calling on Catholic ministry leaders to embrace a more synodal church, a way of listening to the fruits of the Spirit and journeying together as people of faith. Research reveals, however, that many young people have yet to experience this way of being church. This essay proposes a need to transform rather than change ministry with young people to embrace a postmodern curriculum framework rooted in a theology of synodality aimed at cultivating prophetic, transformative, communal experiences of the divine in the world.

**Keywords:** young people; Catholic; synodality; transformation; postmodern curriculum; religious education; accompaniment



## 1. Introduction

There is no shortage of media coverage in the United States of the national surveys and polls documenting the increasing trends in youth and young adult disaffiliation and/or lack of engagement with religious institutions.[1] While such studies highlight important, albeit alarming, trends in young adult[2] religiosity and spirituality, many of these studies not only fail fully describe the lived realities of the young people in our own local faith communities (affiliated or disaffiliated), they also fail to document what working with young people is, often highlighting only problems or worrisome trends.

When faced with problems, it is a common human behavior to search for solutions. Catholic ministry leaders in the United States, many of whom have very limited resources—time being one of the most valuable resources—often gravitate towards ready-made solutions or programs that either come with a promise to help increase church engagement or to reduce disaffiliation. For example, the sociological analysis from the National Study of Youth and Religion (NSYR) theorizes that young people fail to consider morality as an important factor in their decision making and behaviors; that they are becoming moral relativists (Smith 2007–2008; Smith et al. 2011). These researchers believe ministry leaders failed to give young people a robust moral education, leading them to become "morally adrift" (Lamont 2021, pp. 145–164, 149). The quick thinking is that ministers should provide "better" catechesis or more rigorous education on the moral teachings of the Church. Smith et al. conclude that the larger culture and society is to blame for failing to give young adults an adequate moral education (Smith et al. 2011, p. 61). The issue, therefore, is reduced to the problem that young people do not know what constitutes a moral issue, and the solution is to inform or educate them in greater detail on what moral issues are.

This paper argues that the "problem-solution" framework is problematic in three significant ways. Arguing from a white Euro-American perspective, first, the approach to problems stems from inaccuracies with national polls and surveys that create the impression that religious disaffiliation is a static or linear problem to be solved unilaterally rather than a complex, multivalent phenomena unique to each individual and each local community. Second, this approach implicitly reduces religious education and ministry to mere instrumental tools used to solve a perceived statistical problem. Third, and most importantly, the problem–solution approach distances ministers from the lived realities

facing young people in their communities, relegating relationship-building and synodal listening to the periphery of ministry and religious education with young people.

Pope Francis, most notably with the 2021–2023 Vatican Synod "For a Synodal Church: Communion, Participation, and Mission" (synod on synodality), is asking for the Catholic Church to become more synodal in all aspects of church life. According to the Vatican website on the Synod, Synodality:

> denotes the particular style that qualifies the life and mission of the Church, expressing her nature as the People of God journeying together and gathering in assembly, summoned by the Lord Jesus in the power of the Holy Spirit to proclaim the Gospel. Synodality ought to be expressed in the Church's ordinary way of living and working
>
> . . . it is the specific *modus vivendi et operandi* of the Church, the People of God, which reveals and gives substance to her being as communion when all her members journey together, gather in assembly and take an active part in her evangelizing mission (XV Ordinary Council of the Synod of Bishops 2022).

Affirming this definition, Kristin Colberg, one of the four theologians from the United States invited to serve as an assistant to the Synod Bishops on the Vatican Commission, writes:

> From the beginning of his papacy, Francis has called for a synodal church. The word "synod" comes from the Greek, *synodos*, which can literally be rendered as "travelling on a journey together". In essence, the pope is calling for a renewal in the way that the universal church is governed such that the local church—in particular dioceses and bishops conferences—play a more dynamic role in the governance of the universal church" (Colberg 2018, pp. 23–36).

It is my contention that the Church and her leaders in the United States need to critically rethink what it means to understand and practice ministry and religious education with young people. Indeed, the Vatican *Directory for Catechesis* urges catechists, ministers and religious educators to accompany people on the journey of faith by "finding and drawing attention to the signs of God's action already present in the lives of persons . . . " (Pontifical Council for the Promotion of the New Evangelization 2020, #179). Methods for accompanying people in the context of religious education are already present in postmodern education and curriculum theory. The work of scholars such as William Doll, Bell Hooks, William Pinar Dwayne Huebner, Patrick Slattery and Maxine Green, just to name a few, provide the pedagogy and theory to support an education in the faith that has the capacity to bring alive a more synodal way of being church.

This paper uses the method of practical theological reflection to explore an alternative approach to ministry with young people, one not grounded in responses to national statistics, but in a synodal process of listening and accompaniment that unleashes the liberative, prophetic, transformative power which accompanies the process of encountering, listening and discerning—the three adjectives Pope Francis uses to describe synodality (Francis 2021). Theologians begin by observing and describing peoples' lives, their everyday lived experiences. They then ask, what is going on, in what context these observations emerge, and why this might be happening? Similarly, Hispanic/Latinx theologians such as Carmen Nanko-Fernández note how "*lo cotidiano*", or daily living, is a rich source for theological reflection (Nanko-Fernández 2015, p. 15). Kathleen Cahalan writes, "Practical theology turns to multiple sources to critique practice, holding it up to critical scrutiny by a wide range of sources in philosophy and cultural studies" (Cahalan 2011, p. 15). Thus, this essay brings the data on young adult religiosity and spirituality into critical conversation with various disciplines, including sociology, education, religious education and the Christian tradition, and concludes with recommendations to help ministry leaders and religious educators be more responsive to the needs of young Catholic adults.

This research study explores the extent to which national sociological polls and surveys enable ministry leaders to address the lived realities facing Catholic young people in

the United States and offers discussion on the language of change and transformation pertinent to ministry programming. Next, this essay draws insights from the field of religious education, postmodern curriculum theory and the Christian tradition, namely Pope Francis's vision of a more synodal church, to uncover a deeper understanding of ministry and religious education with young people and what it looks like to journey together as the people of God. The essay concludes with insights on how Catholic parish, diocesan, ministry leaders and religious educators can draw inspiration from postmodern curriculum theory and the synodal process to "speak with prophetic criticism and engage in public discourse" so that education with young people becomes "a prophetic enterprise that seeks justice" and where teaching and learning becomes "a moral activity that seeks compassion and understanding" (Slattery 2013, xxi, xx respectively).

## 2. Misinformation of National Surveys

National polls and surveys, such as the National Study of Youth and Religion and Pew Research Center, document the increasing decline of young adults participating in the Church. Most Catholic Churches in the United States affirm these polls as they witness this decline in their faith formation or catechetical programs, ministries, and mass attendance. However, before exploring the relationship between postmodern curriculum development and synodality, it is important to first unpack the nature of national surveys and polling in the United States and its effect on the religious sensibilities of Catholics in the United States.

National reports on the decline in young adult religious affiliation receive a great deal of attention from Catholic pastoral leaders in the United States. For example, after Springtide Research Institute published their sociological study on the increasing rise of loneliness and "epidemic levels of isolation among young people" (Slade 2020), the National Federation for Catholic Youth Ministry (NFCYM) began hosting webinars with Josh Packard, sociologist and executive director of Springtide Research Institute. The organizatoin included links to their research findings in their emails and other communication with their members. Similarly, Bob McCarty, former director of NFCYM, in partnership with St. Mary's press, travels throughout the country presenting for ministry leaders on the findings from the Center for Applied Research in the Apostolate (CARA) and St. Mary's Press on disaffiliation with young people (McCarty and Vitek 2017).

National studies such as these often present ministry leaders with problems that need to be solved. When, for instance, the Pew Research Center survey revealed that only one third of Catholics in the United States believe in the real presence in the Eucharist, archbishops and bishops felt called to respond to this distressing statistic (Pew Research Center 2019a; Smith 2019). The USCCB developed a three-year "Eucharistic Revival" to help Catholics better understand the Eucharist and engage in missionary discipleship (National Eucharistic Revival 2022). The Archdiocese of New Orleans, for example, responded by announcing 2021 as the Year of the Eucharist. To my knowledge, there was no survey or listening sessions conducted with the Catholics in the Archdiocese of New Orleans to assess or better understand their thinking on the Eucharist. The survey data from Pew Research Institute served as the motive to put resources into developing new programs, lectures and retreats, and now a national revival to address this issue—an issue identified by a commercial polling company. Therefore, there will likely be no evidence in local dioceses that these programs and lectures changed peoples' understanding of the Eucharist, given there is no baseline data to compare it to (unless Pew Research Center decides to put out the same poll at the conclusion of the Eucharistic revival). These are just a few examples of how national surveys and polls drive programming and other initiatives in the Catholic Church in the United States.

It is worth noting, however, that there are national Catholic studies and projects such as the *National Dialogue* and the *Fifth (V) National Encuentro* that did host surveys and listening sessions with youth and young adults to explore their experience of church: why young people stay, why they leave, and how ministry leaders and the Church as a whole

can respond (NFCYM 2021; Fifth (V) National Encuentro 2019). Of top concern was the inadequate catechesis or faith formation and the Church's inability to listen to the needs of her young people. However, the issues facing young people have not seen nearly as strong or concerted effort from the USCCB to help dioceses and parishes respond to the concerns young people from around the country raised in relationship to their experiences of church and their faith. The National Dialogue Final Report (NDFR), a publication that documents national and local listening sessions modeled on the synodal process, offered 10 recommendations to help address the needs expressed by Catholic young people (NFCYM 2021, p. 20). To date, there are few, if any, resources from the USCCB directing dioceses and parishes to this report and assisting them with resources to transform faith formation, catechesis, and/or sacramental preparation with young people in the way the Conference is distributing information about the National Eucharistic Revival.

Robert Wuthnow, sociologist of religion, unpacks the history of polling and surveys and the negative impact commercial polling has had on the religiosity of people in the United States in his book *Inventing American Religion* (Wuthnow 2015). Historically, polling only gained public attention through newspapers. News outlets and national media only published polls if they were "newsworthy" (Wuthnow 2015, p. 46). The polling and survey questions were, and still are, designed to catch the public's attention rather than reveal the depth of religiosity held by a particular Christian denomination or local faith community. The latter is the style of research advanced by social scientific scholars in academic institutions. Moreover, polling has long been in the service of U.S. politics as it informs politicians by helping candidates and campaign organizers understand how best to appeal to religious sensibilities. This, according to Wuthnow, was the original function of national polling, which by the 1950s became a multimillion-dollar industry.[3] Furthermore, it is easier for national news media outlets and politicians to discuss or explore broad generalizations for an entire society or country than it is to examine the local nuances of faith and daily living.

Wuthnow also found evidence that revealed only "controversial" or problematic religious statistics were selected for public promotion because they were considered by media outlets to be "newsworthy".[4] His research shows commercial polling organizations, such as Gallup and Barna, are "inventing American religion" in conjunction with the news or other media outlets, that is, the questions the poling companies design and the polls and surveys media outlets find worthy of consideration are the ones that make national attention and thus begin shaping what people think about how this country practice their faith and how they think of religion or religious institutions.

Looking back at the Pew Research study on the Eucharist, the researchers did not survey Catholics to see how they live the Eucharist with their lives, how they experience Christ's presence in the Eucharist, or how the Eucharist draws them into deeper communication with God, self, and others. Rather, they asked a multiple-choice question about transubstantiation, one of the most complex mysteries of the Catholic faith (I have no doubt a similar question will be found in most catechetical or theology textbooks in parishes and schools).[5] Similarly, with the NSYR survey on the moral sensibilities of young people noted above, there was no effort to hear how their answers reflect their sense of religiosity and spirituality, only what the spiritual sensibilities the researchers felt should have come up in their responses, but did not. Furthermore, the researchers failed to examine their responses in light of developmental theory on young adult moral thinking.[6]

In sum, the polling questions themselves fail to reach the depth of peoples' spiritual and religious experiences, and ministry leaders who rely on such data not only look to these incomplete portraits as a source of knowledge about young people, but it also prevents many from going out and listening directly to the young people in their communities. National surveys lead ministry leaders to think they know the problem. What is left is the response to this perceived problem.

As a result, the ministerial responses to national polls and surveys on the religious sensibilities of young people (and adults) too often result in implementing new programs or

curriculum texts aimed at helping young people learn more about the Catholic faith so that they remain affiliated with the Church over time. For example, in 2012, the Pew Research Center published "Nones on the Rise", documenting the alarming decline of religiously affiliated people under the age of 30 (Funk and Smith 2012). In 2021, NDFR, highlighted how most parish sacramental preparation classes were effective in helping young people grow in their faith and remain Catholic, or religiously affiliated. Thus, over the last nine or more years, most parish programs with young people did not help to reduce young adult disaffiliation. As many scholars and practitioners have come to realize, and as the NDFR noted, this approach to ministry with young people is far less effective at helping young people remain active in the Catholic Church (NFCYM 2021, pp. 6, 113). In addition, there is little sustained effort to rethink or transform the traditional approaches to ministry with young adults, only efforts to find new programs or curriculum texts that respond to the perceived problems in young adult religiosity. Responsive approaches to emerging issues can never fully transform ministry with young people, but can only change or pivot what they were already doing towards a new direction.

The next section explores the differences between *change* and *transformation* as they relate to the demands of becoming a more synodal church. While the terms transformation and change are often used interchangeably, and incorrectly so, the pandemic has given many faith communities and ministers a much-needed pause to rethink their purposes, mission, and ministry practices with young people. It is the author's contention that ministry with young adults is ripe for transformation and that postmodern curriculum theory offers a synodal pedagogy to help ministry leaders and religious educators engage in this transformative work. To do this, ministry leaders need to undergo a full transformation in thinking and doing if they are to be effective in engaging the faith lives of young people in a post-pandemic society fragmented by the violence of racial injustice.

### 3. Change or Transformation?

Ministry with young people in Catholic parishes in the United States often takes the form of catechesis or preparation for sacraments, particularly the sacrament of Confirmation, through a classroom-based model of learning. When new statistics emerge, ministry leaders often pivot or change the focus of their instruction, as noted above with the topic of morality, but not their methods or pedagogy. A case-in-point is the way the Archdiocese of New Orleans put out training resources on the Eucharist after the report came from Pew. The formats were lecture-based and didactic in style, and the speakers either held their presentations online or in person for an audience. The same thinking applies to diocesan and parish efforts to respond to the Synod on the New Evangelization by simply changing the name of their ministry programs and curricular material to "Forming Missionary Disciples".[7] In all three examples, ministers did not transform how they engaged people in the faith, rather they simply changed the focus of their educational programs. The ministry formatting remains ministry to or for others.

The *Oxford English Dictionary* defines transformation as "the action of changing in form, shape, or appearance; metamorphosis". In physics, it refers to the "change of form of a substance from solid to liquid, from liquid or solid to gaseous, or the reverse". Similarly, in chemistry, it involves "change of chemical composition, as by replacement of one constituent of a compound by another" (OED Online 2021a, "transformation, n". 2021a). In nature, we see transformation, or a complete metamorphosis, in the life-cycle of a butterfly. Consider the shape and function of a caterpillar with its short thick worm-shaped body crawling on its belly, to then alter its appearance and form completely in its chrysalis to slowly become a winged creature. Its limbs, tissue and organs metamorphosize into a new form that resembles nothing like a crawling caterpillar—it completely and fully transforms into something new, looking nothing like its former self.

The word change, on the other hand, relates "to substitution or exchange". In the *transitive* verb form it means "to substitute one thing for (another); to replace (something) with something else, esp. something which is newer or better; to give up (something) in

order to replace it with something else" ([OED Online 2021b](#), "change, v". 2021b). The butterfly has not substituted one thing for another, nor replaced its body with something else. Likewise, nothing is added on or substituted when water becomes gas; rather, the substance transforms—it becomes something new. In the context of Catholic parish life, most religious education and ministry programs change their focus or topics, but seldom transform their approach or methods.

In the Christian tradition, transformation closely resembles the language of conversion, most notably with the Apostle Paul (once Saul). Indeed, the preparatory document *For a Synodal Church: Communion, Participation, Mission* refers to the synodal process as "synodal conversion" and describes how conversion is a necessary component of evangelization ([XVI Ordinary Council of the Synod of Bishops 2021](#), #2 & #25). Thus, the Synod on synodality is calling for transformation, not change.

The preparatory document likewise describes several significant conversion stories in scripture, for example "in the representation of the 'community scene' that constantly accompanies the journey of evangelization", the way Jesus actively goes out to those in the community, especially "those who are 'separated' from God and those 'abandoned' by the community (the sinners and the poor, in gospel language). Through his words and actions, he offers liberation from evil and conversion to hope".[8]

A second image "refers to the experience of the Spirit in which Peter and the early community recognize the risk of placing unjustified limits on faith sharing" ([XVI Ordinary Council of the Synod of Bishops 2021](#), # 16). It is in the scene from Acts 10 where the authors of the preparatory document note the need for "continuous conversion" to overcome the "fourth actor" in this scene, that of the deceptive ideologies and ways of thinking and doing that come not from the gospel of justice, mercy, love and compassion, but "manifest [themselves] indifferently in the forms of religious rigor, of moral injunction that presents itself as more demanding than that of Jesus . . . ".[9] Transformation and conversion are deeply embedded in the Christian imagination and, as such, offer ministry leaders an exciting, albeit anxious, model for responding to the needs of young Catholics.

In each example, conversion is a transformative experience. Life as these individuals and communities knew it was now completely different because of their experiences with the divine. Such inspiration illustrates the dynamics of living a life of faith in a pluralistic, multivalent, complex, or what scholars call, a "postmodern" society.[10] Conversion is not a single event which occurs once and is over, but an ongoing way of reading the signs of the times and remaining open to the transformation brought about by listening, discernment, and openness to the Holy Spirit. This receptivity and openness, wonder, mystery and conversion or transformation, are precisely what postmodern curriculum theory and the synodal process seek to embody.

Embracing a postmodern approach to curriculum in religious education, as opposed to the modern approach to curriculum operative in most Catholic parishes and schools, will require transformation in how ministry leaders and religious educators understand and engage in faith formation with young people.

## 4. Postmodern Curriculum Theory to Transform Ministry with Young People

Both postmodern curriculum theory and synod-based listening sessions with young people reveal most catechetical instructional methods are not only deficient educational practices, but they also fail to consider the contextuality of teaching and learning. The NDFR suggests such catechetical learning programs aimed at preparing young people to celebrate the sacraments are insufficient in helping young people grow in dialogue with the wider Christian community and fail to engage them in their desire to build up a more just and compassionate world as members of a faith community.

In addition, CARA conducted a national survey on the religiosity of young adults aged 18–35. The researchers found that 60 percent of young adults who identify as Catholic "indicate in some way that they are practicing their faith outside of attending Mass at their parish" ([Gray 2022](#), pp. 3–4). This challenges the statistics from other polling companies

such as Pew Research Center, which base their reports of the increasing decline of religiosity predominantly on how often people attend worship or mass (Pew Research Center 2019b). CARA found that of the 60% of young adults who said they are Catholic, the majority attended some sort of religious programming for youth as adolescents. In the Catholic context, such programming is largely carried out as sacramental preparation. As these young Catholics move towards young adulthood, the majority express feeling they can grow in their faith outside of traditional parish programming and the parish itself. This is an important reality for church leaders to unpack.

As religious educators, our implicit and explicit understanding of curriculum and pedagogy frames our efforts in teaching and learning in ministry with young people. Modern curriculum theories, those most associated with standardized testing and rigid curriculum frameworks, draw insight from advances in science and technology and from European Enlightenment philosophers such as René Descartes and his emphasis on ordered reason and observable truths (Slattery 2013). This way of approaching curriculum design stems from what is known as the Tyler Rationale. Ralph Tyler wrote a series of books on curriculum in 1949 which served to provide teachers with a structured approach to the day-to-day classroom setting by articulating content objectives, designing learning activates and assessing the process of learning in the student (Tyler 1949). Little thought was given to the purposes of education holistically, only to the day-to-day teaching methods.

Patrick Slattery, scholar in curriculum theory, provides a concise summary of the paradigm shift that moved the modern curriculum methods toward postmodern curriculum theories. He states, "we have been conditioned" by modernism and the modern curriculum "to believe that our goals, objectives, lesson plans, and educational outcomes must all be *measurable* and *behaviorally observable* in order to be valid" (Slattery 2013, p. 24). We see evidence of this when ministry leaders respond to national polls documenting the decline in morality in young people, as noted above. The sociologists who documented this problem offered a measurable solution: provide young adults with more moral education. Slattery goes on to state:

> I have met very few teachers who actually believe this philosophy of education. However, the majority who do not ascribe to this educational ideology–rooted in scientific management and the Tylerian Rationale–have allowed themselves to be conditioned to behave as though they do. Postmodernism challenges educators to explore a worldview that envisions schooling through a different lens of indeterminacy, aesthetics, autobiography, intuition, eclecticism, and mystery.[11]

A postmodern approach offers a way to transform ministry with young adults away from the "problem-solution" response and toward a more, holistic, organic interdependent style of accompaniment. It is a process that encourages ministry leaders and young adults journey with one another, to grow in faith together, so that more young adults experience the parish as a place where they feel they belong and are spirituality nourished.

The espoused methodology in the modern curriculum views knowledge as the search for truth based on what one can prove or, according to William Doll, "whatever is true, factual, real is discovered, not created" (Doll 1993, p. 31). The legacy and influence of the Tylerian model of education is evidenced in the national push for standardized testing in both public and Catholic schools United States (Slattery 2013, pp. 17–73).[12] Indeed, the USCCB *Doctrinal Elements of a Curriculum Framework for the Development of Catechetical Materials for Young People of High School Age* and its adapted iteration for use in parish religious education programs illustrate this standardization of religious education and approach to curriculum development.

Many students in the United States are, as a result, accustomed to "showing what they know" through standardized multiple-choice tests. There is nothing more prescribed or less open to ambiguity and wonder than choosing the right "answer" from four or five pre-determined choices. It is also noteworthy that national commercial polling companies use a similar method to assess the way people think about or understand a topic. Meanwhile, young adults are asking for ministry leaders to accompany and mentor them through the

complexity that comes with living a life of faith. Both CARA and the NDFR reported that young adults do not want ready-made answers to complex questions. Most young adults report wanting dialogue partners, people who are authentic and ready to listen and who will "accompany them as they seek out a relationship with God and to discern their future" (NFCYM 2021, p. 108; Kramarek 2022, p. 35).

"Curriculum development postmodern era", according to Slattery, "demands that we find a way around the hegemonic forces and institutional obstacles that limit our knowledge, reinforce our prejudices, and disconnect us from the global community" (Slattery 2013, p. 36). Drawing insight from Michael O'Malley, Slattery describes how the dominant educational model in use today centers on a Euro-American style of education and curriculum that fosters an "ethic of exclusion" structured on a divided life. Slattery states this dichotomy leads to "the conscious absence of soul from education" which, in turn, "limits the efficacy of the pedagogical project and actually creates conditions in which social ills—anxiety, racism, poverty, exclusion—flourish".[13] Thus, didactic modes of catechesis, devoid of communal engagement, listening and dialogue, risk falling into this "ethic of exclusion". In this way, the implicit curriculum of most parish catechetical programs works against the method and practice of synodality.

Today, most young adults report growing in their faith outside traditional parish structures (programs, mass, ministries). It stands to reason that parish programming with youth needs to help establish a firmer connection to their faith if they hope to grow as young adults within the context of the parish. As it stands, most young adults associate parish programming with sacramental preparation, which in most parishes is a classroom-based standardized model of teaching and learning. "Many youth", according to the NDFR, "engaged in the Sacraments simply because of obligation or because we "told them to.' They were looking for meaning and purpose, both of which the Church and Sacraments can offer, but often did not connect them to those religious and spiritual experiences" (NFCYM 2021, p. 113). Young adults want more from the parish if they are to grow in their faith. They want spaces where they can engage in true, authentic dialogue about the things most relevant to living a life of faith.

In sum, there is a disconnect between what most young adults say they want from their churches and the way ministry leaders are engaging them in their youth and as young adults today. Curricular methods that frame programming with young people need to create spaces where young adults feel a sense of belonging, where they are listened to and recognized for their agency (Deck 2022, p. 82). Thus, young adults are asking for ministry leaders to "reimagine faith formation ... away from a classroom model and towards more relevant learning models featuring mentorship, small groups, accompaniment, faith sharing, and authentic witness" (NFCYM 2021, p. 111). Postmodern approaches to ministry and religious education, will, according to Slattery, encourage the community "to become mentors and guides who will inspire students to seek wisdom and understanding as a part of a community of learners ... fellow travelers on the lifelong journey of learning" (Slattery 2013, p. 115). In essence, they want the Church and her programming to better model "a culture of synodality" (NFCYM 2021, p. 115).

## 5. Synodality and the National Dialogue

In 2018, a group of national pastoral ministry leaders and organizations supported by the United States Conference of Catholic Bishops (USCCB) sought to respond to the increasing trends in youth and young adult disaffiliation and disengagement in parish life but also to recognize the ways in which pastoral ministry positively impacts the lives of young people. This initiative, called The National Dialogue on Catholic Pastoral Ministry with Youth and Young Adults, was "a collaborative and synodal experience of the Church in the United States, is the answer to Pope Francis' call for accompaniment, conversation, and dialogue".[14]

To accomplish their goals, the core team and national leadership network identified three key outcomes from this synodal process. "The collaborating organizations looked

to this project to bring unity to the ministry field, to engage youth and young adults and ministry leaders in meaningful dialogue, and to mobilize the Church to integrate and implement the key insights from this process, the Synod, and the V Encuentro".[15] I was honored and enlightened to be a part of this process; to experience first-hand what synodality can look like. Ministry leaders submitted data from their own listening sessions in the form of dialogues or open-ended conversations in their respective contexts from listening to parents of young people, affiliated or disaffiliated youth or young adults, and ministry leaders themselves.[16] I held my own listening sessions with ministry leaders and parents who then held listening sessions with affiliated young adults.

This data, compiled in the NDFR,[17] confirmed other national surveys on the decline in church membership and youth and young adult disaffiliation but also added important nuances to these statistics that only local listening sessions can reveal. For instance, in using open-ended questions rather than pre-selected multiple choice answers, there were some issues, such as the clergy sexual abuse crisis that simply did not come up as much in these sessions as often as other surveys would presuppose (Jones 2019). In addition, NDFR found that while most young people have had less than positive experiences of catechesis or faith formation, there were still many who did receive fruitful experiences in youth ministry programming and they requested more opportunities to gather young people for "conferences, retreat type activities, leadership formation, and service opportunities".[18] This brief discussion on the findings illustrates the complexity of ministry with young people that seldom surfaces in other national surveys and polls, such as Pew Research Institute, Gallup, or Springtide Research Institute.

While such statistics help us understand general patterns of human behavior, they cannot tell a person what is in the hearts and minds of the people they minister with. What is more alarming, in my view, is the way in which ministry leaders rely on national polls and surveys to tell them about the people in their own communities, rather than going out to them and listening to the people themselves.

I saw this process unfold when I facilitated a listening session with ministry leaders and stakeholders at a Catholic school who wanted to better engage the young adult alumni in the community. In getting to know one another before the formal listening session began, participants shared with me a variety of approaches they felt would help young adults grow in their faith. I heard a lot of "this way is the best", or "I know this style is the way to grow discipleship . . . ". However, when the listening session started and I asked individuals in the group to describe young people who they actually knew (rather than their assumptions about young people), their comments changed. They noted how many young people did not feel welcomed at church, they want meaningful relationships, a sense of community, and a space to ask questions without judgement. When we went over the summary of their responses in our next session, the group agreed that their job as ministry leaders and, for some of them, as parents, was to grow these ministerial and characteristics with young people if they are ever going to become or remain active in their faith.

When we returned the next time to process the first dialogue session, one participant noted how important the first listening session was and pointed out how their group was in a deeper relationship now, that what they shared was powerful and sacred. From the experience of sharing their lived experiences and what was in their hearts and minds authentically, as it related to young adults, their relationship to the Church and their faith in what became a safe, yet brave space, this gathering of ministry leaders and stakeholders became a small Christian community.

This is one reason why Pope Francis is calling on ministry leaders to become a more synodal church. Francis, therefore, devoted an entire synod, 2021–2023 For a Synodal Church: Communion, Participation, and Mission, to this theme as a way of educating and training people in this way of being church. Indeed, during the last Synod on Young People, the Faith, and Vocational Discernment, the pope specifically states that "youth [and young adult] ministry has to be synodal . . . " (Francis 2019, # 206). He continues:

> It should involve a "journeying together" that values "the charisms that the Spirit bestows in accordance with the vocation and role of each of the Church's members, through a process of co-responsibility . . . Motivated by this spirit, we can move towards a participatory and co-responsible Church, one capable of appreciating its own rich variety, gratefully accepting the contributions of the lay faithful, including young people and women, consecrated persons, as well as groups, associations and movements. No one should be excluded or exclude themselves

> In this way, by learning from one another, we can better reflect that wonderful multi-faceted reality that Christ's Church is meant to be. She will be able to attract young people, for her unity is not monolithic, but rather a network of varied gifts that the Spirit ceaselessly pours out upon her, renewing her and lifting her up from her poverty.[19]

It is my contention that the only way to transform parish religious education and ministry programming around a more synodal style is to adopt postmodern educational approaches and methods. The modern curriculum framework and experience of most Catholic young people, as noted above, fails to cultivate dialogue, complexity, and methods of accompaniment. Postmodern curriculum helps ministry leaders develop more emergent models in pastoral ministry and religious education rather than prescribed. Such approaches will begin first by inviting young people to dialogue, to share with ministers in a safe, yet brave space, how the Church has helped them or brought them joy; how the church has let them down; and what we can do together to move forward. All listening sessions should be conducted with ground rules for sharing, such as the ones developed by Parker Palmer called "The Circle of Trust®"[20].

Synodality is an emergent process, not a prescriptive one. Unlike a textbook or prescribed program, we will not know the outcome in advance. Synodality, like postmodern curriculum theory, is about listening to others regarding their experience of church and discerning with the Holy Spirit how to respond faithfully to the hopes, dreams, needs and concerns of the people in our parishes. Thus, the Synod on Young People and the subsequent synod on synodality invite ministry leaders and religious educators to transform not only how they minister with young people, but to re-envision what it means to be church. Business as usual no longer works.

## 6. Conclusions

In my experience, most ministry leaders see national polls and surveys, look at their empty pews and low or declining numbers at youth or young adult events, and start looking for quick solutions. They begin looking for new programs to fix these issues, a "silver bullet" that helps them stop the tide of disaffiliation, indifference to the Church, or lack of engagement in parish life. Religious educators hear their bishop, pastor, or even the pope talk about evangelization as a way of forming missionary disciples, so they go out and find the new textbook or catechetical or program that adopts this new language and begin using it with young people. Then, when another synod, papal document, bishops' statement, or societal issue arises, they either pivot or are asked to pivot their programming again to address a new concern. All the while, they have not engaged in any efforts to transform the way they think about and do church. Very few go out and listen to the people directly involved in their concerns, relying instead on generalizable statistics and ready-made solutions.

Furthermore, these ways of engaging in youth and young adult ministry are not only reactive in orientation, but they fail to move beyond just small changes in programming. These approaches fail to connect with the lived realities of the people in our ministries and faith communities. This is all to say that synodality has yet to become a way of being church in most parishes in the United States.

What is needed for the Church today, in a country afflicted by the sins of racial injustice perpetuated by white supremacy and laboring to emerge from the COVID-19 pandemic, is real transformation—not change. Ministry leaders need to develop a completely new way of being church, one, as this paper argues, that emerges through the process of synodality. My own experiences with the National Dialogue, research on postmodern curriculum development, and the the Synod on Synodality give me hope that pastoral ministers can enact the transformation needed to accompany young people in life.

Postmodern curriculum theory helps us see the deficiencies of the modern schooling and instructional methods found in most traditional catechetical classrooms or other programs with young people and exposes them for their limits, for their inability to cultivate prophetic, transformative, communal experiences of the divine in the world. Postmodern curriculum theory helps us transform our understanding of religious education and ministry programming; synodality gives us a process by which we might do this.

Synodality is the way forward in ministry with young people. Developing more synodal church helps us develop a posture that looks to every moment for an opportunity to see and experience glimpses of God in others and in ourselves. It is a radical worldview open to wonder, to being surprised by just how compassionate and loving God is. Post-modern ministry with young people transforms our way of being church by propelling the internal disposition of pastoral ministers to "go out to others, seek those who have fallen away, stand at the crossroads, and welcome the outcast" (Francis 2013, #24). Rather than changing the content of our programming so that young people better understand the Church's teachings, a postmodern curriculum framework based on synodality invites ministry leaders to form relationships with young people, to create spaces where they feel they can voice what is in their hearts, question and grow spiritually, all the while being attuned to God's presence in our midst and in each encounter we have.

When ministry leaders engage in deep listening with young people, they are drawn into relationships with one another. When we hear what is in a young person's heart and in their mind, ministers become responsible bearers of their story. When we teach through autobiography, they become bearers of *our* story. From here, ministry leaders start the "journey together" as the synodal process continues to unfold as the minister and young people discern the fruits of their dialogue and to act responsively. In this way, education and ministry with young people becomes "a prophetic enterprise that seeks justice", and where teaching and learning becomes "a moral activity that seeks compassion and understanding" (Slattery 2013, p. xxi.) Together, they are poised to stand in solidarity with one another and as a strong, meaningful, small, Christian community, ready to engage in the transformation of society.

**Funding:** This research received no external funding.

**Institutional Review Board Statement:** Not applicable.

**Informed Consent Statement:** Not applicable.

**Data Availability Statement:** Not applicable.

**Conflicts of Interest:** The author declares no conflict of interest.

## Notes

1 Some examples include (Sciupac 2020; Newport 2018; Packard et al. 2020).
2 This essay follows the United States Conference of Catholic Bishops (USCCB), who define the young adult years as between the ages of 18 and 39, whereas youth or adolescents are considered those of ages 13–18.
3 Ibid., 64.
4 Ibid., 120.
5 The question reads: Which of the following best describes Catholic teaching about the bread and wine used for Communion? The bread and wine . . . 1. Actually become the body and blood of Jesus Christ; 2. Are symbols of the body and blood of Jesus Christ; 3. Not sure. For a closer look at the implications of this question, see (Gray 2019).
6 For more on this, see (Lamont 2021)
7 For a full discussion on this, see (Lamont 2020).
8 Ibid, # 16 & # 17.
9 (Ibid., #21).
10 See for example, (Smith 2006; Lakeland 1997) and the extensive bibliography there.
11 Ibid., p. 24.
12 For a full discussion on how the modern curriculum is operative in schools, see (Slattery 2013, pp. 17–73).
13 (Ibid., p. 111).
14 The National Dialogue, *Home*.
15 NFCYM 2021, p. 11. "The Synod" refers to the 2018 Synod on Young People, the Faith and Vocational Discernment.
16 Ibid., 64
17 Ibid.
18 Ibid., 84.
19 Ibid., #206–207.
20 (See Palmer 2018); The National Dialogue: Host a Conversation; The Way of Proceeding for Facilitators and Participants.

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
