# Peer review of "From Change to Transformation: Living Synodality in Ministry with Young Adults"

_religions, doi:10.3390/rel14030314_

Round 1

Reviewer 1 Report

To start with, I would like to congratulate the author of this very important matter. The author approached the subject matter in an interesting way. Hence the work is both "unique and original". The author talks about synodality and the’path’ of change to transformation. 

Unfortunately, the context is not precise and it is not always clear if this vision applies only to the author’s home country  or to other parts of the Catholic world.  The author needs to state exactly what the synodality way is according to the Catholic Church and the suggestions offered by the Holy Father. His Holiness does not see synodality as building a dialogue with post-modern culture. They are on two different levels.

It would be better if the author had prepared a proper bibliography of specialist authors to whom further addition reference could be made.

In my opinion the paper is of interest but I have strong reservations about this work being published.

Author Response

Thank you for  your review. I made additional notes about the context (Catholic, United States), throughout the essay. I added a definition of synodality with examples on how to become a more listening church. From my research and understanding, Postmodern curriculum theory (not culture) embraces elements of synodality that will help ministry leaders and religious educators move away from their current programming models and more toward an emergent model of Church that the Pope Francis and the synodal process embraces.

I included a bibliography. Please see revisions using track changes attached

Reviewer 2 Report

I agreed to do a review of this article because the title seemed to me interesting and very up to date. I was mainly expecting suggestions on how to implement synodality in todays youth ministry. Meanwhile, the author describes things that are commonly known, various generalities. He devotes a lot of space to criticizing the Tylerian model of education. He is right here, but this is not new in studies and it does not contribute much to research. There is also a controversial opinion like "more rigorous education on the moral teaching". I hope that this is not the author's proposal for the pastoral care of today's youth, because that would have the opposite effect. I find it very unfortunate that there is little there on the practical application of synodality because the author, as he writes, is a member of "The National Dialogue on Catholic Pastoral Ministry with Youth and Young Adults." On the topic signaled in the title, he wrote only a few sentences. They are also well-known opinions that contribute little to science and pastoral work.
I suggest that the author add a significant part of the article and provide suggestions for concrete actions to implement synodality in dialogue with young people. If this happens, the article can be very interesting and useful.

Author Response

The Tylerian model of education is not well known in Catholic pastoral ministry or religious education.  The Tyler Rationale is the operative (albeit implicit) model in contemporary Catholic parish catechetical programs and needs to evolve towards more postmodern curriculum theories and practices if synodality is to take hold.

The author did not advance the notion that "more rigorous education on the moral teaching". She wrote that this advice is advanced by Christian Smith et all from the National Study of Youth and Religion. My essay argues the opposite, that their approach is insufficient.

I added more practical ways to engage in synodality from my experience with the National Dialogue.

Please see the attached revisions using Track Changes.

Reviewer 3 Report

Thank you for the opportunity to read and review this article.

In general, the argument that we should have a more synodally-informed approach to catechetical instruction is well made. And overall, I'm persuaded. The authors muster a significant amount of theological and educational theories in order to suggest that a new way is not only possible but also warranted.

The conceptual argument presented in this manuscript could be improved if the authors considered the following points:

1) Sections 2, 3, and 4 are well presented as independent units, but I think more "connective tissue" could be created across these three sections. As they currently stand, each section establishes a premise (in section 2, that opinion polls are an insufficient way to track the spirituality of youth; in section 3, that change and transformation will look different if we embrace a truly synodal approach; in section 4, that current educational trends in the U.S. will not provide appropriate guidance to a truly synodal approach to catechesis). But I think by the time the authors reach sections 5 and 6, it is incumbent on the authors to have these arguments more tightly knit together.

2) I think section 4 requires more work and potentially a significant revision. Too many educational theories are too quickly summarized. Granted, this is a brief manuscript and the authors are very clearly drawing on these educational theories in order to make a point about the catechesis of youth in our current moment. But I worry that this relatively cursory overview of these themes means that the authors have sacrificed analytical precision in order to be more concise.

In general, though, I think the argument and the writing in this manuscript hold together well.

Author Response

Thank you for the very helpful review. I revised the manuscript to show more closely how the themes from Sections 2, 3, and 4 connect.

I worked on section 4 to make this more robust.

Please see the revisions attached using Track Changes

Round 2

Reviewer 1 Report

I would like to thank the author for making changes to the text, which in my opinion are a very good answer to the allegations made in the review. Thank you that the author did not treat the comments in a personal way, but approached the issue professionally. Science always demands substantive and methodological precision. 

The solution of the problem is well set out in the conclusion of the work. The bibliography is well choosen and those authors are of good standing. It raises no objections in terms of content and methodology. The bibliography and statistical methods have been applied correctly.

The footnotes could be given the traditional form (Arabic numerals, not Roman numerals).

The pattern of proceedings presented in this way has led to conclusions that are consistent with the evidence and reasoning applied. In my opinion the paper is of good value and therefore I would suggest publishing it.

Reviewer 2 Report

In my opinion, this article is more pastoral than academic. However, it can be accepted for publication because it is one of the first articles on the subject and can be a starting point for further discussion.
It also meets the objectives of the special issue on 'Catholic Education and Pope Francis' Dream for a Synodal Church'.
It must also be said that the author has made significant improvements to the original version and now the article seems better.